# Contrastive Learning from Pairwise Measurements

**Yi Chen**[†]    **Zhuoran Yang**[‡]    **Yuchen Xie**[†]    **Zhaoran Wang**[†]
[†]Northwestern University    [‡]Princeton University
{yichen2016, ycxie}@u.northwestern.edu
zy6@princeton.edu    zhaoran.wang@northwestern.edu

## Abstract

Learning from pairwise measurements naturally arises from many applications, such as rank aggregation, ordinal embedding, and crowdsourcing. However, most existing models and algorithms are susceptible to potential model misspecification. In this paper, we study a semiparametric model where the pairwise measurements follow a natural exponential family distribution with an unknown base measure. Such a semiparametric model includes various popular parametric models, such as the Bradley-Terry-Luce model and the paired cardinal model, as special cases. To estimate this semiparametric model without specifying the base measure, we propose a data augmentation technique to create virtual examples, which enables us to define a contrastive estimator. In particular, we prove that such a contrastive estimator is invariant to model misspecification within the natural exponential family, and moreover, attains the optimal statistical rate of convergence up to a logarithmic factor. We provide numerical experiments to corroborate our theory.

## 1   Introduction

Learning from pairwise measurements is a fundamental problem which arises in many real-world scenarios, such as the customers' preferences among a set of alternative products, the performance of athletes in a tournament, and the allocation of resources among a set of competing alternatives. In this paper, we consider the setting where the users perform evaluations on alternative items based on pairwise comparisons. That is, the data are generated by the preferences of the users on item pairs randomly selected from the set of alternatives. Various models and estimators have been proposed to study the pairwise measurements problem. For example, the Bradley-Terry-Luce Model (Bradley and Terry, 1952) uses Bernoulli distributions with parameters quantifying the users' preferences on alternative items to explain the users' choices. In this case, the maximum likelihood approach can be used to estimate the model parameters. In spite of the empirical success in many applications, most of the existing models and the corresponding estimators are based on specific distributions of pairwise measurements, whose correctness hinges on that the models are correctly specified. However, in practice, without the knowledge of the true model, applying these methods might incur huge estimation errors. To alleviate this problem, in this paper, we consider a semiparametric model for pairwise comparison which incorporates a broad class of distributions of the pairwise measurements, and contains various popular parametric models, such as the Bradley-Terry-Luce model and the paired cardinal model (Gopalan et al., 2014), as special cases. Specifically, we assume that the pairwise measurements follow a natural exponential family distribution that is parametrized by a underlying score matrix. The underlying score matrix quantifies the users' evaluations on different items. In addition, to characterize the similarities between users and items, we further assume that underlying

score matrix has a latent representation, that is, it can be decomposed as the multiplication of two low-dimensional matrices, which represent the properties of the items and the preferences of the users on these properties, respectively. In contrast to other works, we do not specify the base measure of the natural exponential family distribution, but instead treat it as a nuisance parameter. Thus, our model is semiparametric and we are interested in estimating the underlying score matrix, which corresponds to the parametric component of our model. By considering the parameter estimation problem for such a semiparametric model class, we achieve invariance to model misspecification within the natural exponential family.

For our semiparametric model, it is challenging to estimate the model parameters, since the likelihood function of the pairwise measurements is intractable due to the existence of the nonparametric nuisance parameter. To overcome such difficulty, we use a data augmentation technique to create virtual examples, which enables us to define a contrastive loss function. Then we construct a contrastive estimator based on these virtual examples, instead of the original data. In specific, we define the virtual dataset by taking differences between all pairs of the data points, i.e., a virtual example is constructed by two pairwise comparisons. These virtual examples are used to construct a contrastive loss function that can be interpreted as the likelihood for the rank statistics of paired pairwise measurements. Interestingly, this contrastive loss function does not depend on the base measure of the exponential family distribution and is only a function of the score matrix of interest. Moreover, the true score matrix is the unique minimizer of the population version of this loss function. Therefore, with the same spirit as maximum likelihood estimation, we define the contrastive estimator as the minimizer of the contrastive loss function. Furthermore, we apply nuclear norm regularization to capture the low-rank structure of the underlying score matrix. Since our contrastive estimator does not rely on the the base measure, it remains valid no matter what the true distribution of pairwise measurements is, so long as the distribution is within the natural exponential distribution family.

Our contrastive estimator is based on the virtual examples, which are the pairwise differences of the pairwise measurements and can be viewed as comparison of comparisons. Although it seems that we recover the underly score matrix in an indirect and inefficient way, as we will show in the main theorem, our contrastive estimator is consistent and nearly achieves the optimal rate of convergence. It is worth mentioning that one challenge of the convergence analysis is that the virtual examples are not independent due to the comparisons between the pairwise measurements. To circumvent this problem, we use a novel decoupling trick to obtain sharp concentration rates. Moreover, we also provide numerical simulations to corroborate the main theorem, which demonstrate that our statistical rate of convergence is tight asymptotically.

**Related Work:** The problem of learning from pairwise measurements dates back to the Bradley-Terry-Luce Model for the rank aggregation problem, which is first proposed by Bradley and Terry (1952). Following the Bradley-Terry-Luce Model, several extensions with increasing complexity and descriptive power have been raised, such as the Plackett-Luce Model (Luce, 2005; Plackett, 1975) and the Multinomial Logit Model (Negahban et al., 2017). See, for example, Negahban et al. (2012, 2016) and the references therein for more recent developments. Various algorithms and estimators are also proposed to estimate these models (Soufiani et al., 2013; Guiver and Snelson, 2009; Soufiani et al., 2013), and many recent works are devoted to understanding their statistical properties (Rajkumar and Agarwal, 2014; Khetan and Oh, 2016). In particular, Lu and Negahban (2015) characterizes the statistical error of the maximum likelihood estimator of the Bradley-Terry-Luce Model and Negahban et al. (2017) conduct a finite-sample analysis for the Multinomial Logit Model. Despite the satisfactory progress, all of these works focus on choices and ranks, while in many cases, the pairwise measurements data are not categorical and might contain more information than choices and ranks. Several recent work addresses this issue, for example, the Paired Cardinal Model uses Gaussian distribution to characterize the pairwise measurements (Gopalan et al., 2014), and Shah et al. (2015) uses Poisson distribution to model the click-through rate in recommendation systems. Compared with these works, our work is novel in that we do not specify the underlying distribution of the pairwise measurements and our estimator is invariant to model misspecification when the true distribution varies within a broad class of distributions.

Our work is also closely related to another line of study on the matrix completion with noise (Keshavan et al., 2010). The original matrix completion problem aims at recovering the missing entries of a partially-observed matrix. See, for example, Candès and Recht (2009) and the references therein. More recently, various work focuses on the problems where the observations are contaminated by noise. For example, Keshavan et al. (2010) investigates the case of Gaussian noise; Cao and Xie (2015) deals with the problem where the observed data follow Poisson distributions parametrized by the matrix entries, and Gunasekar et al. (2014); Lafond (2015) further extends the distribution of noise to exponential family. For more related works in this area, see, for example, Keshavan et al. (2010); Klopp et al. (2017); Sambasivan and Haupt (2018). Compared with these models, our work recovers a matrix from partially-observed pairwise differences of entries without specifying the distribution of noise. Our data augmentation technique, which constructs a valid estimator regardless of model misspecification, can also be potentially useful in the task of matrix completion with noise. Finally, the idea of constructing a nuisance-free loss function based on data augmentation is studied in Liang and Qin (2000); Chan (2012); Diao et al. (2012); Ning et al. (2017); Yang et al. (2014) for semiparametric regression models. Compared with these work, although we adopt the same loss function, due to the pairwise structure, analyzing the convergence rate of our estimator requires a careful treatment of the dependence between the virtual samples. We apply a novel decoupling method to obtain sharp concentration results.

**Main Contribution:** In summary, the contribution of this paper is twofold: (i) For our semiparametric model, we propose a novel contrastive estimator that is invariant to model misspecification when the true distribution of pairwise measurements belongs to the family of natural exponential family. (ii) We show that our contrastive estimator is consistent and achieves the nearly optimal statistical rate of convergence. This enables us to learn the model efficiently based on finite pairwise measurements data. Moreover, in the convergence analysis, we apply a decoupling trick to obtain sharp concentration for dependent random variables, which might be of independent interest.

**Organization:** We first introduce the background and our model assumptions in §2. Then we define our contrastive estimator and explain why it works in §3. We lay out the main theorem in §4, which characterizes the rate of convergence of the contrastive estimator. Several numerical results are presented in §5 to demonstrate our main theorem. We conclude our paper in §6. Due to space constraint, we defer all of the detailed proofs to §A and §B of the appendix.

**Notation:** Throughout this paper, for a positive integer $n$, we use $[n]$ to denote the set $\{1, \ldots, n\}$. For two matrices $A, B \in \mathbb{R}^{d_1 \times d_2}$, we use $\langle A, B \rangle$ to denote their inner product, which is defined as $\text{Tr}(A^\top B)$. For a matrix $X$, we also use $\|X\|_\infty$, $\|X\|_2$, $\|X\|_*$, and $\|X\|_\text{F}$ to denote its infinity, spectral, nuclear, and Frobenius norms, respectively.

## 2 Background

In this section, we first introduce a unified statistical model of representation learning from pairwise measurements and then present several examples.

We assume that there are $d_1$ users and $d_2$ items, where $d_1$ and $d_2$ are positive integers. A generalized comparison by a user is generated in three stages. First, a user with label $j$ is selected uniformly at random from the $d_1$ users. Then user $j$ compares a pair of items, namely, $k$ and $\ell$, which are drawn uniformly at random from the $\binom{d_2}{2}$ possible pairs of items. In the following, we use a triple $\mu = (j, k, \ell)$ to denote the comparison between items $k$ and $\ell$ made by user $j$. Finally, we observe a generalized comparison $y \in \mathbb{R}$ corresponding to the triple $\mu$, which is specified as follows.

Let $\Theta^* \in \mathbb{R}^{d_1 \times d_2}$ be the underlying score matrix, where $\Theta^*_{j,k}$ denotes user $j$'s score of item $k$. We assume that $\Theta^*$ has a latent representation where each entry $\Theta^*_{j,k}$ is determined by the inner product of two vectors $u^{(j)} \in \mathbb{R}^r$ and $v^{(k)} \in \mathbb{R}^r$, which are the vector representations of user $j$ and item $k$, respectively. In specific, the entries of $v^{(k)}$ represent the properties of item $k$, for example, its quality and price, while the entries of $u^{(j)}$ encode the preferences of user $j$, that is, how much it weighs different properties of an item. Correspondingly, the score of an

item given by a user is determined by the user-specific weighted average of the item's properties. Equivalently, $\Theta^*$ takes the form of $\Theta^* = UV^\top$, where $U = (u^{(1)}, u^{(2)}, \ldots, u^{(d_1)})^\top \in \mathbb{R}^{d_1 \times r}$ and $V = (v^{(1)}, v^{(2)}, \ldots, v^{(d_2)})^\top \in \mathbb{R}^{d_2 \times r}$ contain the vector representations of users and items as their rows, and $r \ll \min\{d_1, d_2\}$ is the dimension of vector representations. Such a latent representation assumption is motivated by the fact that, in practice, the scores given by different users may be close due to the similarities between users or items. For example, when choosing from various genres of movies, users of the same age or gender are more likely to share similar preferences.

We assume that the generalized comparison $y$ is the realization of a random variable $Y$. Given the triple $\mu = (j, k, \ell)$, the density of $Y$ takes the form of a natural exponential family distribution with natural parameter $\Theta^*_{j,k} - \Theta^*_{j,\ell}$, that is,

$$p_\mu(y; \Theta^*) = \exp\{y(\Theta^*_{j,k} - \Theta^*_{j,\ell}) - b_\mu(\Theta^*) + \log f(y)\}, \qquad (2.1)$$

where $f$ is the base measure function and

$$b_\mu(\Theta^*) = \log\left(\int_{\mathcal{Y}} \exp\{y(\Theta^*_{j,k} - \Theta^*_{j,\ell}) + \log f(y)\}\mathrm{d}y\right)$$

is the normalization constant, which depends on $\Theta^*$ and $f$. Here $\mathcal{Y}$ denotes the domain of $Y$. Note that shifting each row of $\Theta^*$ with constants leads to the same distribution of $\{y_i\}_{i \in [n]}$, which makes $\Theta^*$ non-identifiable. Hence, for sake of the identifiability of true latent representation, we assume that $\sum_{k=1}^{d_2} \Theta^*_{j,k} = 0$ for all $j \in [d_1]$. We also assume that $\|\Theta^*\|_\infty \leq \alpha$ in order to controls how "distributed" the learned latent representation is. It captures the dynamic range of the underlying score matrix as well.

Note that different base measure functions $f$ lead to specific distributions of $Y$, for example, Bernoulli, Gaussian, binomial, and other common distributions. In particular, the parameterization in (2.1) generalizes several existing models of ranking, crowdsourcing, and ordinal embedding. Here are some examples that correspond to different base measures in (2.1).

**Bradley-Terry-Luce Model:** Let the base measure be the constant function $f(y) = 1$ and $Y \in \{0, 1\}$ be a binary random variable. Then the model in (2.1) reduces to

$$\mathbb{P}(Y = 1) = \frac{\exp(\Theta^*_{j,k})}{\exp(\Theta^*_{j,k}) + \exp(\Theta^*_{j,\ell})}, \quad \mathbb{P}(Y = 0) = \frac{\exp(\Theta^*_{j,\ell})}{\exp(\Theta^*_{j,k}) + \exp(\Theta^*_{j,\ell})},$$

which is known as the Bradley-Terry-Luce model (Bradley and Terry, 1952) if we interpret $Y = 1$ as that user $j$ prefers item $k$ to item $\ell$ and $Y = 0$ as the opposite.

**Paired Cardinal Model:** Let the base measure be the exponential function $f(y) = 1/\sqrt{2\pi} \cdot \exp(-x^2/2)$ and $Y$ be a continuous random variable. Then the model in (2.1) reduces to $Y = \Theta^*_{j,k} - \Theta^*_{j,\ell} + \epsilon$, where $\epsilon \sim N(0, 1)$. This model is known as the paired cardinal model in Shah et al. (2015).

**Poisson Arrival Model:** Let the base measure be $f(y) = 1/y!$ and $Y$ be a nonnegative integer-valued random variable. Then given the triple $\mu = (j, k, \ell)$, the response $Y$ follows Poisson distribution with mean parameter $\Theta^*_{j,k} - \Theta^*_{j,\ell}$. The Poisson arrival model is commonly used to characterize the click-through rate in recommendation systems (Gopalan et al., 2014). This model is also related to the Poisson matrix completion model (Cao and Xie, 2015), which finds applications in communications and signal processing.

**Ordinal Embedding Model:** The ordinal embedding problem (Jain et al., 2016) aims to recover the (relative) positions of $d$ points by comparing their pairwise distances. In specific, for $d$ points $x^{(1)}, \ldots, x^{(d)} \in \mathbb{R}^r$, their Euclidean distance matrix $D \in \mathbb{R}^{d \times d}$ is defined by $D_{j,k} = \|x^{(j)} - x^{(k)}\|_2^2$. Given a triple $\mu = (j, k, \ell)$, we observe a realization of the binary random variable $Y \in \{0, 1\}$ with

$$P(Y = 1) = \phi(D_{j,\ell} - D_{j,k}), \quad P(Y = 0) = 1 - \phi(D_{j,\ell} - D_{j,k}), \qquad (2.2)$$

where $\phi : \mathbb{R} \to [0, 1]$ is a known link function. Our goal is to recover the distance matrix $D$ based on $n$ such observations. When $\phi(z)$ is the logistic link $\exp(z)/(1 + \exp(z))$, the ordinal embedding model in (2.2) coincides with the model in (2.1) with $f(y) = 1$ and $D = -\Theta^* + C$, for any absolute

constant $C$. If we further restrict $x^{(1)}, \ldots, x^{(d)}$ to be on the unit sphere, then by additionally setting $d = d_1 = d_2$, $u^{(j)} = v^{(j)} = \sqrt{2} \cdot x^{(j)}$, and $C = 2$, we obtain the same model as (2.1) with the latent representation $\Theta^* = UV^\top$, since we have $D_{j,k} = \|x^{(j)} - x^{(k)}\|_2^2 = 2 - 2x^{(j)\top}x^{(k)}$.

The maximum likelihood estimation of all the above models requires us to specify the distribution of $Y$ in advance, which is determined by the base measure function $f$ in (2.1). However, the exact distribution of $Y$ is often unknown in practice and can be misspecified, which results in inaccurate estimation of the latent representation. To alleviate this issue, we treat the base measure function $f$ as a nuisance parameter, and introduce a new contrastive estimator of the latent representation based on virtual examples, which unifies the estimation of all the above models.

# 3 Contrastive Representation Learning

In this section, we present a data augmentation technique that creates virtual examples, which is inspired by the likelihood ratio approach for seimiparametric regression models studied in Liang and Qin (2000); Chan (2012); Diao et al. (2012); Ning et al. (2017); Yang et al. (2014). Through the rank-order decomposition, such virtual examples enable us to construct a simple and unified estimator, especially a new contrastive loss function, to learn the latent representation without specifying the exact distribution of pairwise measurements.

In the sequel, we denote by $\{\mu_i\}_{i \in [n]}$ the observed indices of measurements, which are $n$ independent copies of $\mu$. Here $\mu_i = (j(i), k(i), \ell(i))$ is the $i$-th observed triple and $n$ is the sample size. We use $y_i$ to denote the $i$-th response of pairwise measurement corresponding to $\mu_i$, which is a realization of $Y$ following the distribution in (2.1). Our goal is to estimate the underlying score matrix $\Theta^*$ based on $\{(y_i, \mu_i)\}_{i \in [n]}$. Before we introduce the estimator, we define the virtual dataset

$$\mathcal{V} = \left\{ \left(y_p - y_q, M^{(p)} - M^{(q)}\right) : p, q \in [n], p < q \right\}. \tag{3.1}$$

Here $M^{(i)}$ ($i \in [n]$) is the measurement matrix corresponding to the triple $\mu_i = (j(i), k(i), \ell(i))$, which is defined as $M^{(i)} = E_{j(i),k(i)} - E_{j(i),\ell(i)}$. Recall that $E_{s,t}$ is the $(s,t)$-th matrix canonical basis of $\mathbb{R}^{d_1 \times d_2}$.

**Contrastive Estimator:** Our estimator takes the following form

$$\widehat{\Theta} \in \underset{\Theta \in \mathcal{C}(\alpha)}{\mathrm{argmin}}\, L_n(\Theta; \mathcal{V}) + \lambda R(\Theta). \tag{3.2}$$

Here $L_n$ denotes the contrastive loss function, which depends on $\{(\mu_i, y_i)\}_{i \in [n]}$ through the virtual dataset $\mathcal{V}$ defined in (3.1), and $R$ is the penalty function. In (3.2), $\lambda > 0$ denotes the regularization parameter and $\mathcal{C}(\alpha)$ is the parameter space of $\Theta$ with $\alpha > 0$, which is specified as

$$\mathcal{C}(\alpha) = \left\{\Theta \in \mathbb{R}^{d_1 \times d_2} : \|\Theta\|_\infty \leq \alpha,\ \textstyle\sum_{k=1}^{d_2}\Theta_{j,k} = 0,\ j \in [d_1]\right\}. \tag{3.3}$$

The definition of parameter spaced $\mathcal{C}(\alpha)$ corresponds to our assumptions on true parameter $\Theta^*$, which is specified in §2. In detail, the contrastive loss function $L_n$ takes the following form

$$L_n(\Theta; \mathcal{V}) = \frac{2}{n(n-1)} \sum_{p,q \in [n], p<q} \log\left(1 + \exp\left\{-(y_p - y_q)\langle M^{(p)} - M^{(q)}, \Theta\rangle\right\}\right). \tag{3.4}$$

Intuitively, $L_n(\Theta; \mathcal{V})$ is the logistic loss on the virtual dataset $\mathcal{V}$ with size $|\mathcal{V}| = \binom{n}{2}$, where we use the virtual covariate $M^{(p)} - M^{(q)}$ to predict the virtual response $y_p - y_q$. We name $L_n$ as the contrastive loss function because it is based on the difference between pairs of data points. Meanwhile, recall that $\Theta^*$ has the latent representation $\Theta^* = UV^\top$ with $U \in \mathbb{R}^{d_1 \times r}$, $V \in \mathbb{R}^{d_2 \times r}$, and $r \ll \min\{d_1, d_2\}$. In (3.2), we use the nuclear norm penalty $R(\Theta) = \|\Theta\|_*$ to enforce the low-rank structure of $\widehat{\Theta}$.

**Why It Works?** In the following, we explain how the contrastive loss function $L_n$ in (3.4) based on the virtual dataset $\mathcal{V}$ enables us to recover the latent representation without knowing the exact distribution in (2.1), which is determined by the base measure function $f$. The key idea is the rank-order decomposition. In detail, we consider a pair of independent observations $(y_p, \mu_p)$ and $(y_q, \mu_q)$, which can be decomposed into the order statistics $o_{p,q} = (\min\{y_p, y_q\}, \max\{y_p, y_q\})$ and

rank statistics $r_{p,q} = \mathbb{1}\{y_p \le y_q\}$. Let $o_{p,q}$ and $r_{p,q}$ be the realizations of random variables $O$ and $R$. Then we have

$$
\begin{aligned}
\mathbb{P}(R = r_{p,q} \,|\, O = o_{p,q}; \Theta) &= \frac{\mathbb{P}(R = r_{p,q}, O = o_{p,q}; \Theta)}{\mathbb{P}(R = r_{p,q}, O = o_{p,q}; \Theta) + \mathbb{P}(R = r_{q,p}, O = o_{q,p}; \Theta)} \\
&= \frac{p_{\mu_p}(y_p; \Theta) \cdot p_{\mu_q}(y_q; \Theta)}{p_{\mu_p}(y_p; \Theta) \cdot p_{\mu_q}(y_q; \Theta) + p_{\mu_p}(y_q; \Theta) \cdot p_{\mu_q}(y_p; \Theta)},
\end{aligned}
\tag{3.5}
$$

where $p_\mu(y; \Theta)$ is defined in (2.1). Note that we have

$$
p_{\mu_p}(y_p; \Theta) \cdot p_{\mu_q}(y_q; \Theta) = \exp\Big\{ y^{(p)} \big\langle M^{(p)}, \Theta \big\rangle + y^{(q)} \big\langle M^{(q)}, \Theta \big\rangle \Big\} \cdot \xi_{p,q}(f, \Theta),
$$

$$
p_{\mu_p}(y_q; \Theta) \cdot p_{\mu_q}(y_p; \Theta) = \exp\Big\{ y^{(q)} \big\langle M^{(p)}, \Theta \big\rangle + y^{(p)} \big\langle M^{(q)}, \Theta \big\rangle \Big\} \cdot \xi_{q,p}(f, \Theta),
$$

where by (2.1) we have

$$
\xi_{p,q}(f, \Theta) = \xi_{q,p}(f, \Theta) = \exp\Big\{ -b_{\mu_p}(\Theta) - b_{\mu_q}(\Theta) + \log f(y_p) + \log f(y_q) \Big\}.
\tag{3.6}
$$

Note that the unknown base measure function $f$ affects the conditional probability in (3.5) only through $\xi_{p,q}(f, \Theta) = \xi_{q,p}(f, \Theta)$ defined in (B.15), which are cancelled in the numerator and denominator. As a result, we obtain

$$
\mathbb{P}(R = r_{p,q} \,|\, O = o_{p,q}; \Theta) = 1 \Big/ \Big( 1 + \exp\Big\{ -(y_p - y_q)\big\langle M^{(p)} - M^{(q)}, \Theta \big\rangle \Big\} \Big),
\tag{3.7}
$$

which does not depend on the base measure function $f$. Following the maximum likelihood principal, we take the negative logarithm of product of $\mathbb{P}(R = r_{p,q} \,|\, O = o_{p,q}; \Theta)$ across all possible pairs $(y_p, \mu_p)$ and $(y_q, \mu_q)$, which gives the contrastive loss function $L_n$ in (3.4).

# 4 Main Results

In this section, we establish the nonasymptotic rate of convergence for the contrastive estimator defined in (3.2). Before we present the main theorem, we lay out the following regularity conditions. Recall that, throughout this paper, we use $C$, $C'$, and $C''$ to denote absolute constants whose values may vary from line to line.

The first regularity condition concerns the statistical model defined in (2.1). Recall that the density of a natural exponential family distribution with parameter $\theta$ takes the form

$$
p(x; \theta) = \exp\big\{ x\theta - b(\theta) + \log f(x) \big\}.
\tag{4.1}
$$

Here $f$ is the base measure function and $b(\theta)$ is the normalization constant, which depends on $\theta$ and $f$. Given the measurement triple $\mu = (j, k, \ell)$, the statistical model in (2.1) is a special case of (4.1) with parameter $\theta = \langle M, \Theta^* \rangle$, where $M = E_{j,k} - E_{j,\ell}$. We consider independent random variables $X, X' \in \mathcal{X}$, whose density takes the form of (4.1) with different parameters $\theta$ and $\theta'$ but the same base measure function $f$. We define $m(\theta, \theta') : \mathbb{R} \times \mathbb{R} \to \mathbb{R}$ as

$$
m_\alpha(\theta, \theta') = \mathbb{E}\Big[ (X - X')^2 \cdot \psi\big(8\alpha|X - X'|\big) \Big], \quad \text{where} \ \ \psi(z) = \exp(z)\big/\big(1 + \exp(z)\big)^2.
\tag{4.2}
$$

Here $\alpha$ is defined in (3.3) and the expectation is taken with respect to the joint distribution of $X$ and $X'$, which depends on $\theta$ and $\theta'$. The following regularity condition is on the continuity of $m_\alpha(\theta, \theta')$.

**Assumption 4.1** (Continuity). We assume that $m_\alpha(\theta, \theta')$ is a continuous function.

One can show that, for all natural exponential family distributions on bounded domains, Assumption 4.1 holds. Meanwhile, for common exponential family distributions on unbounded domains, such as Gaussian, exponential, and Poisson distributions, one can verify that Assumption 4.1 also holds.

Informally speaking, Assumption 4.1 ensures that the population version of the contrastive loss function defined in (3.4) is strongly convex. In other words, it guarantees that the eigenvalues of the population Hessian matrices at all parameters are lower bounded away from zero, which implies the identifiability of the true parameter. In specific, let $(Y, M)$ and $(Y', M')$ be two independent pairwise measurements as specified in §3, which are generated by the statistical model in (2.1) with the true parameter $\Theta^* \in \mathcal{C}(\alpha)$. Here recall that the parameter space $\mathcal{C}(\alpha)$ is defined in (3.3). For

$\Theta \in \mathcal{C}(\alpha)$, the population version of the loss function takes the form

$$L(\Theta) = \mathbb{E}\Big[\log\Big(1 + \exp\{-(Y - Y') \cdot \langle M - M', \Theta \rangle\}\Big)\Big], \tag{4.3}$$

where the expectation is taken with respect to the joint distribution of $(Y, M)$ and $(Y', M')$. Formally, we have the following proposition.

**Proposition 4.2.** Under Assumption 4.1, for all $\Theta \in \mathcal{C}(\alpha)$ and $\Delta \in \mathbb{R}^{d_1 \times d_2}$ such that $\sum_{k \in [d_2]} \Delta_{j,k} = 0$ for all $j \in [d_1]$, there exists a constant $\beta > 0$, which depends on $\alpha$, such that

$$1/2 \cdot \mathrm{vec}(\Delta)^\top \nabla^2 L(\Theta) \, \mathrm{vec}(\Delta) \geq 1/2 \cdot \mathbb{E}\Big[m_\alpha\big(\langle M, \Theta^* \rangle, \langle M', \Theta^* \rangle\big) \cdot \langle M - M', \Delta \rangle^2\Big]$$

$$\geq \beta/(d_1 d_2) \cdot \|\Delta\|_F^2, \tag{4.4}$$

where $\nabla^2$ denotes taking the second-order derivative with respect to $\mathrm{vec}(\Theta) \in \mathbb{R}^{d_1 d_2}$, and the expectation is taken with respect to the joint distribution of $M$ and $M'$.

*Proof.* See §B.7 for a detailed proof. $\qquad\square$

As shown in Proposition 4.2, the first term in (4.4) is the quadratic form associated with the population Hessian matrix evaluated at $\Theta$. Such a quadratic form is universally lower bounded by the second term, which does not depend on $\Theta$, and is further lower bounded by $\beta/(d_1 d_2) \cdot \|\Delta\|_F^2$. In other words, under the conditions of Proposition 4.2, the population loss function defined in (4.3) is strongly convex.

Recall that $\{y_i\}_{i \in [n]}$ consists of $n$ independent observed responses of the pairwise measurements. The following regularity condition is on the boundedness of $y_i$ ($i \in [n]$).

**Assumption 4.3** (Boundedness). There exists $\nu > 0$ such that $\max_{i \in [n]} |y_i| \leq \nu$.

The regularity condition on boundedness is used to simplify our subsequent discussion. Even when the response of pairwise measurement has unbounded support, Assumption 4.3 holds with high probability for a properly chosen $\nu$, which possibly depends on the sample size $n$. For example, when $Y$ is a sub-Gaussian random variable, by union bound it suffices to set $\nu = C\sqrt{\log n}$ for a sufficiently large absolute constant $C > 0$. Furthermore, we remark that we this assumption is required only for establishing concentration results. However, our contrastive estimator is still valid even when this assumption is violated. Moreover, in this case, we can use a truncation argument to establish a similar rate of convergence.

We are now ready to present the main theorem. Recall that $\widehat{\Theta}$ is the contrastive estimator defined in (3.2) and $\Theta^*$ is the true score matrix with rank $r$. The following theorem upper bounds the difference between $\widehat{\Theta}$ and $\Theta^*$ in Frobenius norm.

**Theorem 4.4.** We assume that Assumptions 4.1 and 4.3 hold and $n \leq C/(\alpha^4 \beta^2) \cdot (d_1 d_2)^{2/3} \cdot \log(d_1 + d_2)$. Let the regularization parameter in (3.2) be

$$\lambda = C' \nu \cdot \big(1/\sqrt{n \cdot \min\{d_1, d_2\}} + 1/n\big) \cdot \log(d_1 + d_2).$$

Then the following inequality holds with probability at least $1 - C/(d_1 + d_2)$,

$$\frac{1}{\sqrt{d_1 d_2}} \cdot \big\|\widehat{\Theta} - \Theta^*\big\|_F \leq C'' \max\{1/(\alpha\beta), \nu/\beta\} \cdot \left(\sqrt{\frac{r \max\{d_1, d_2\}}{n}} + \frac{\sqrt{r d_1 d_2}}{n}\right) \cdot \log(d_1 + d_2). \tag{4.5}$$

Here $\beta$ is the constant specified in Proposition 4.2.

*Proof Sketch.* See §A of the appendix for a detailed proof. Our proof consists of two key steps. First, by combining concentration inequalities with a peeling argument, we establish the strong convexity of the contrastive loss function defined in (3.4) within a restricted subset of the parameter space. The major challenge is that the contrastive loss function is a sample average over the virtual dataset $\mathcal{V}$ defined in (3.1), in which the data points are dependent. Such dependency prohibits us from applying

standard concentration inequalities. To overcome this difficulty, we decouple the dependent sample average into a hierarchical average that involves permutations of data points. This allows us to eliminate the dependency and establish concentration inequalities via moment generating functions. See §B.1 for the proof of the strong convexity and see §B.3 for the details of the decoupling method. Second, we upper bound the gradient of the contrastive loss function at the true parameter in spectral norm. The major challenge is that the aforementioned decoupling of dependent sample average prohibits us from applying the standard matrix Bernstein inequality. Instead, we resort to Taylor's expansion of the moment generating function and apply the matrix Bernstein inequality to each order of moment. See §B.2 for more details. Finally, by combing the two steps with a geometric analysis, we establish the rate of convergence for the contrastive estimator. □

Theorem 4.4 establishes the statistical rate of convergence of the contrastive estimator $\widehat{\Theta}$ defined in (3.2). As specified by the parameter space defined in (3.3), roughly speaking, $\|\Theta^*\|_{\mathrm{F}}$ scales with $\sqrt{d_1 d_2}$. Hence, on the left-hand side of (4.5), we scale the estimation error in Frobenius norm with $1/\sqrt{d_1 d_2}$ for proper normalization. Our result shows that, with a suitable choice of the regularization parameter $\lambda$, the rescaled estimation error is of the order $\sqrt{r \max\{d_1, d_2\}/n} + \sqrt{rd_1 d_2}/n$ up to a logarithmic factor. In particular, for $n \gg \min\{d_1, d_2\}$, the leading term of the error is $\sqrt{r \max\{d_1, d_2\}/n}$. Note that, since the $\Theta^* \in \mathbb{R}^{d_1 \times d_2}$ is rank-$r$, the total number of unknown parameters in $\Theta^*$ if of the order $r \max\{d_1, d_2\}$. Hence, our result in (4.5) achieves the optimal rate of convergence for estimating $\Theta^*$ up to a logarithmic factor.

It is worth mentioning that the upper bound of the sample size $n$ required by Theorem 4.4 may be not necessary. It arises due to the peeling argument in the proof, which also appears in previous literature (Lu and Negahban, 2015; Negahban et al., 2017), but can be eliminated by a more delicate analysis. However, for the simplicity of discussion, we do not pursue this direction in this paper.

## 5 Numerical Experiments

We lay out the simulation results in this section and demonstrate the accuracy of the contrastive estimator $\widehat{\Theta}$ stated in Theorem 4.4. Throughout our numerical experiments, for the sake of simplicity, we always use an equal number of users and items, that is, we set $d_1 = d_2 = d$. We also use Bernoulli distribution as the distribution of the pairwise measurements, which is specified in (2.1). For a fixed $d$ and $r$, in order to generate an underlying score matrix $\Theta^* \in \mathbb{R}^{d \times d}$ of rank at most $r$, we first generate two matrices $U \in \mathbb{R}^{d \times r}$ and $V \in \mathbb{R}^{r \times d}$, whose entries are independently and identically distributed standard normal random variables. Note that the sum of each row in the underly score matrix needs to be zero for the sake of identifiability. Therefore, for each row of $V$, we subtract the sample mean of that row from each entry, that is, we compute $\widetilde{V}$ as $\widetilde{V}_{ij} = V_{ij} - 1/d \cdot \sum_{l=1}^{d} V_{il}$. Finally, we set $\Theta^* = U\widetilde{V}$. Then, we generate $n$ pairwise measurements following the mechanism specified in §2. Note that the optimization problem in (3.2) is a convex optimization with nuclear norm regularization, for which we solve by using the proximal gradient method (Cai et al., 2010). For the regularization parameter $\lambda$, we set $\lambda$ in (3.2) as $0.5 \cdot 1/\sqrt{nd} \cdot \log(2d)$, as suggested in Theorem 4.4.

Recall that Theorem 4.4 implies that with high probability, the dimensional-rescaled estimation error $1/d \cdot \|\widehat{\Theta} - \Theta^*\|_{\mathrm{F}}$ scales of order $1/\sqrt{n}$ when $r$ and $d$ are fixed, and increases with $r$ as well as $d$. To support the conclusion of Theorem 4.4, we perform two sets of experiments, whose results are plotted in Figure 1. In the first set, the rank is fixed at $r = 2$ and different dimensions $d \in \{5, 10, 15, 20\}$ are tested. In the second set, the dimension is fixed at $d = 20$, and different ranks $r \in \{2, 4, 6, 8\}$ are tested. In both cases, we run simulations with sample size $n \in \{200, 400, \ldots, 3000\}$. In Figure 1. (a)-(b), we plot the rescaled estimation error $1/d \cdot \|\widehat{\Theta} - \Theta^*\|_{\mathrm{F}}$ against $1/\sqrt{n}$, for different choices of $(r, d)$. As shown in the figures, for fixed choices $(r, d)$, the points approximately lies on a straight line that goes through the origin, implying that the rescaled estimation errors is approximately proportional to $1/\sqrt{n}$. Moreover, its slope increases as $r$ or $d$ increases, which is consistent with Theorem 4.4.

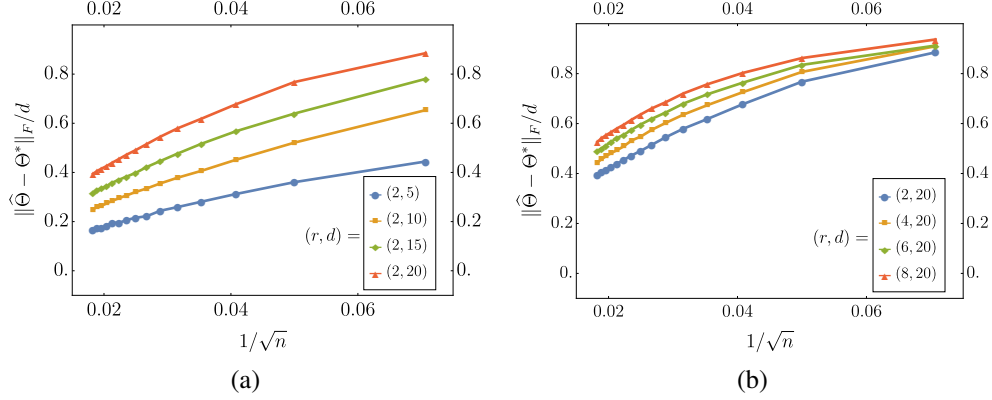

Figure 1: Plot of rescaled estimation error $1/d \cdot \|\widehat{\Theta} - \Theta^*\|_F$ against $1/\sqrt{n}$, where $n$ is the sample size. In (a), the rank $r$ is fixed at $r = 2$, and the dimension $d$ is taken as $d = 5, 10, 15, 20$. In (b), the dimension $d$ is fixed at $d = 20$, and the rank is taken as $r = 2, 4, 6, 8$.

## 6    Conclusion

In this paper, we consider the problem of learning from pairwise measurements where the distribution of measurements follows from a natural exponential family distribution, where the base measure is assumed unknown. For such a semiparaemtric model, we use a data augmentation technique to construct a contrastive estimator that is consistent and achieves nearly the optimal statistical rate of convergence. Moreover, our estimating procedure is agnostic to the base measure of the exponential family distribution, thus achieving the invariance to model misspecification within the natural exponential family.

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
