[Reviews · NeurIPS 2018]

Reviewer 1



The paper considers a family of statistical models for pairwise measurements, in which we observe scores that come from triplets of the form (user, item1, item2). The problem is to estimate the underlying score function of user-item pair, which is assumed to be low-rank, and the observation to be generated by a natural exponential family. The authors address the problem from a novel angle as a semiparametric estimation problem, which avoids the need to specify the log-partition function in the natural exponential family. A penalized contrastive estimator is derived and its consistency and error bound are analyzed. Numerical simulation supports the rate established in the bound. I find this a nice addition to the learning from pairwise comparison literature. I recommend accept. Detailed Comments: 1. Equation (3.2) (3.4) effectively reduces the problem to a logistic loss matrix sensing problem with a specific randomly generated sensing matrix. Can a similar robustness to model misspecification be stated for other sensing matrices and other link function? 2. The exponential family distribution of p(y|Theta) really needs to be a natural exp. family. In general, one still needs to specify the model class in order to reparameterize a given exp family to a natural one. In that sense, the semiparametric view is not as strong as it seems. 3. Assumption 4.3 is somewhat strong as it rules out Gaussian, Poisson and many others. Perhaps it suffices to use sub-gaussian or sub-exponential?

Reviewer 2



The paper considers the problem of learning from pairwise measurement. The model is in the form of an exponential family distribution, and includes some well-known models as its special cases. The proposed contrastive estimator is based on a transformation of the original data, thus eliminates the requirement of specifying the base measure function for the model. The statistical rate of convergence is proved for the estimator showing that it achieves the optimal rate of convergence up to a log factor. The paper is nicely written, and has a good discussion comparing the work with learning from pairwise measurement and matrix completion. The removal of the base measure assumption is very interesting, making the estimator attractive and robust when specifying the parametric model itself is not easy. The experiment part is not adequate. Some real world experiments are needed to support the usefulness of the estimator, while the paper only has simulation results. Even the simulation data are rather small, in terms of matrix size.

Reviewer 3



This paper considers problem of learning pairwise measurements when the underlying distribution follows a natural exponential family distribution. Specifically, the paper creates virtual examples and optimize the contrastive loss over them. From Eq. (3.4), my understanding is: if y_p and y_q are closer, the corresponding loss should be smaller. If we have a multi-class problem where y_p > y_k > y_q, hence y_p – y_q > y_k – y_q; for this scenario, is the loss function Eq. (3.4) still reasonable? The paper focuses on optimization problems with simple parameter space C(\alpha) in Eq. (3.3) as a constraint. Will the derivation still hold if complex constraints are involved? On Line 44, the paper considers to use nuclear norm as a regularization to ensure low-rank structure. Is this a necessary part for the derivation?